# The Role of Strategic Emotional Intelligence in Predicting Adolescents’ Academic Achievement: Possible Interplays with Verbal Intelligence and Personality

**DOI:** 10.3390/ijerph182413166

**Published:** 2021-12-14

**Authors:** Zorana Jolić Marjanović, Ana Altaras Dimitrijević, Sonja Protić, José M. Mestre

**Affiliations:** 1Department of Psychology, Faculty of Philosophy, University of Belgrade, 11000 Belgrade, Serbia; aaltaras@f.bg.ac.rs; 2Institute for Educational Psychology “Rosa & David Katz”, Faculty of Philosophy, University of Rostock, 18051 Rostock, Germany; 3Institute for Criminological and Sociological Research, 11000 Belgrade, Serbia; sonja.milojevic@iksi.ac.rs; 4International Psychoanalytic University, 10555 Berlin, Germany; 5University Institute of Social and Sustainable Development (INDESS), University of Cádiz, 11405 Jerez de la Frontera, Spain; 6Department of Psychology, University of Cádiz, 11519 Puerto Real, Spain

**Keywords:** strategic emotional intelligence, emotion understanding, emotion management, verbal intelligence, Big Five, academic achievement, adolescents

## Abstract

As recent meta-analyses confirmed that emotional intelligence (EI), particularly strategic EI, adjoins intelligence and personality in predicting academic achievement, we explored possible arrangements in which these predictors affect the given outcome in adolescents. Three models, with versions including either overall strategic EI or its branches, were considered: (a) a mediation model, whereby strategic EI partially mediates the effects of verbal intelligence (VI) and personality on achievement; the branch-level version assumed that emotion understanding affects achievement in a cascade via emotion management; (b) a direct effects model, with strategic EI/branches placed alongside VI and personality as another independent predictor of achievement; and (c) a moderation model, whereby personality moderates the effects of VI and strategic EI/branches on achievement. We tested these models in a sample of 227 students (*M* = 16.50 years) and found that both the mediation and the direct effects model with overall strategic EI fit the data; there was no support for a cascade within strategic EI, nor for the assumption that personality merely moderates the effects of abilities on achievement. Principally, strategic EI both mediated the effects of VI and openness, and independently predicted academic achievement, and it did so through emotion understanding directly, “skipping” emotion management.

## 1. Predicting Academic Achievement from Individual Dispositions

Academic achievement typically refers to performance outcomes in intellectual domains covered within instructional environments at different academic levels [1,2]. From a psychological perspective, it is essential not only because it determines prospective educational and vocational opportunities (e.g., university enrolment) [1,2] but also because it is a central indicator of positive psychological functioning in children and adolescents [3]. Within this group, higher academic achievement is positively related to subjective well-being [4,5], life satisfaction [6,7], and happiness [8]. In addition, academic success can profoundly affect one’s self-perception, as it contributes to higher self-efficacy [9] and the formation of a positive academic self-concept [10]. Considering the role of academic achievement in children’s and adolescents’ optimal development and functioning, understanding its correlates and predictors becomes “important both theoretically and practically, warranting continued scholarly pursuit” [11] (p. 33).

Academic achievement is a complex outcome variable, known to depend on various factors. While situational variables certainly play a role, they leave much of the variance in this outcome unexplained, implying that students’ individual dispositions may greatly decide how well they fare at school [1,12]. Traditionally, intelligence and personality have been considered as the most influential dispositional predictors of academic achievement [1]. More recently, researchers have realized that students’ emotions at school also strongly affect their motivation, learning strategies, and ultimately their performance [13]; thus, emotional intelligence (EI), i.e., the ability to cognitively process and deal with these emotions, has come into the foreground as another individual disposition implicated in academic success [14].

### 1.1. Intelligence

That intelligence should play an essential role in school attainments has been a general premise since the beginning of intelligence testing; in fact, these tests were initially designed to assess children’s capacity to master the curriculum. To this day, intelligence measures build their validity on the successful prediction of academic achievement [15]. There is a common understanding among intelligence researchers that this individual trait captures one’s “ability to understand complex ideas, to adapt effectively to the environment, to learn from experience, to engage in various forms of reasoning, to overcome obstacles by taking thought” [16] (p.77). Thus, intelligence is bound to predict achievements in various fields, but particularly those in the academic realm, which draw heavily on abstract reasoning and learning.

Indeed, early narrative reviews of the association between scholastic achievement and intelligence found them to be substantially related, with a mean correlation of 0.50 [16,17,18]. The hitherto most comprehensive meta-analytic study [19], which included 240 independent samples of students from elementary to high school reported a corrected correlation of *ρ* = 0.54 between *g* and school grades, thus corroborating earlier findings. In addition, the respective meta-analysis also yielded important insights into the moderators of this relationship. First, out of the three types of intelligence tests considered in moderator analyses, a higher population correlation was observed for verbal and mixed measures (*ρ* = 0.53 and 0.60, respectively) than for nonverbal ones (*ρ* = 0.44). This finding was interpreted with regard to the importance of verbal skills for successful classroom interaction, as well as for taking oral and written exams. Second, moderator analyses confirmed that the population correlation increased from elementary (*ρ* = 0.45) to middle (*ρ* = 0.54) to high school (*ρ* = 0.58), suggesting that lack of ability could more easily be compensated for by hard work in lower grades, but that the importance of intelligence increases as students face more challenging study contents [19]. From this point, it becomes crucial to understand how different dispositional traits combine and interact to produce positive academic outcomes.

### 1.2. Personality

Interestingly, it was the very leaders of the intelligence-testing movement who were among the first to acknowledge that performance depends on other personal factors besides intelligence [20,21]. That is why much of the research on academic achievement has also been devoted to establishing how students’ characteristic patterns of experiencing and acting in certain situations—in short, their personality traits—relate to performance in school. Nevertheless, the findings of these studies remained quite scattered and inconsistent until the appearance of broad factorial models of personality that provided a common framework for such studies [22]. The widely accepted Five-Factor Model of personality [23] was particularly potent in this regard. Within this model, personality is described in terms of five broad dimensions, commonly referred to as the Big Five: neuroticism (N), extraversion (E), openness (O), agreeableness (A), and conscientiousness (C). 

To explore the relationship between the Big Five and academic performance, Poropat [22] performed a large meta-analysis on 47 to 138 samples (depending on the trait) of students from elementary school to university. To assess the meaningfulness of the obtained correlations, which in such large samples (*N* = 58,522–70,926) were statistically significant even when smaller than 0.10, Cohen’s *d* was calculated, thus revealing a medium-sized effect of C (*d* = 0.46) and small effects of O (*d* = 0.24) and A (*d* = 0.14) on academic achievement; the effects of N and E were minor. Additional analyses corroborated the conclusion that, among the Big Five, C is the most stable predictor of academic performance. Its correlation with the criterion increased after controlling for intelligence, and remained unaffected by education level and student age, leading to the conclusion that “future considerations of individual differences with respect to academic performance will need to consider not only the *g* factor of intelligence, but also the *w* [willingness] factor of Conscientiousness” [22] (p. 334).

### 1.3. Emotional Intelligence

The concept of EI was introduced by Salovey and Mayer [24] to acknowledge individual differences in the ability to reason about emotions and to use them to enhance thought. More precisely, EI is the “ability to perceive emotions, to access and generate emotions so as to assist thought, to understand emotions and emotional knowledge, and to reflectively regulate emotions so as to promote emotional and intellectual growth” [25] (p. 5). According to this definition, EI comprises four “branches,” which are thought to be hierarchically ordered, ranging from emotion perception (EP), as the most basic branch, through using emotions, to the more complex branches of emotion understanding (EU) and management (EM) [25,26]. The two lower branches constitute the experiential area, while the two higher ones form the strategic area of EI.

Soon after EI was proposed and established as a meaningful construct (see [26] for an overview of corroborating findings), research attention also turned to its role in promoting achievement at school. While it is acknowledged that EI’s primary domain of relevance is in predicting (inter)personal rather than performance outcomes, it was also convincingly argued that EI abilities might be implicated in succeeding at school. For example, Ivcevic and Brackett [11] hypothesized that school outcomes are influenced by two distinct types of self-regulation dispositions: typical performance traits, such as C, and maximum performance attributes, such as the ability to understand and manage one’s emotions (as defined within the EI construct). Both were presumed to aid achievement-related behaviors and to do so independently of each other. For example, a tendency to work hard should promote the completion of challenging and long-term tasks, but so should the ability to modulate unpleasant emotions that often accompany such tasks (e.g., frustration or anxiety) and bring about maladaptive reactions (e.g., procrastination or rumination). Similarly, Lopes et al. argued that the ability to regulate emotions contributes to academic achievement by “sustaining the motivation to pursue learning or mastery goals in the face of frustration or self-doubt” [27] (p. 716). Another possible mechanism through which EI could aid school performance is by facilitating positive peer and teacher interactions, and the expression of school-appropriate behaviors, hence providing the necessary social conditions for successful learning [27,28]. Finally, as a third possibility, it was also suggested that EI, particularly the EU branch, might be directly involved in mastering academic content in the language arts and humanities, i.e., in subjects that require an understanding of people and their emotions [14].

In line with these theoretical proposals, numerous studies have indeed found EI to be positively associated with school performance. The results of these studies were recently meta-analyzed by two independent research groups, leading to the common conclusion that EI significantly correlates with academic achievement, with estimates of the population correlation being *ρ* = 0.24 [14] and $\bar{Z}$ = 0.31 [29]. MacCann et al.’s [14] study further found strategic EI—i.e., emotion understanding and management—to have a significantly larger effect on academic achievement than the two lower EI branches. Moreover, EI and its strategic branches incrementally predicted academic performance over intelligence and personality, and a relative weights analysis showed EU and EM to be the second-best predictors of academic achievement, coming after intelligence but before C.

## 2. The Interplay between Intelligence, Personality, and EI as Predictors of Academic Achievement

Hitherto, the main research question concerning the role of EI in academic outcomes has been whether EI can indeed predict school performance, and if so, whether its contribution goes beyond what is already explained by individual differences in intelligence and personality. Now that this question has been resolved, a new one comes to the foreground: How does EI interact with intelligence and personality to predict academic achievement? In the following passages, we consider several possible models of interplay.

### 2.1. Mediation and “Cascading” Models

First, it is conceivable that EI mediates the relationship between intelligence and personality, on the one side, and academic performance on the other (Figure 1a). This model is reminiscent of the one proposed by Joseph and Newman [30] in their attempt to establish how the same trio of predictors relates to job performance. More precisely, Joseph and Newman’s model also assumes a partial mediation effect, with intelligence and personality predicting the criterion both directly and through EI. As both intelligence and personality were more recently shown to explain nontrivial variance in EI [31], there is further justification for the hypothesized mediation.

A distinctive feature of Joseph and Newman’s model, lending it the name “the cascading model of EI,” is that the domain of EI is represented by three branches—EP, EU, and EM—forming a causal sequence from EP as the most distal to EM as the most proximal predictor of job performance (i.e., EP→EU→EM) [30]. The results of both Joseph and Newman’s meta-analysis and of a more recent study by Nguyen et al. [32] yielded support for the proposed cascading effects within EI, showing EU to largely [30] or fully [32] mediate the relationship between EP and EM. In view of this, but given the current focus on predicting academic achievement (rather than job performance), we considered an analogous yet somewhat more parsimonious model: here, the EI variable would be narrowed down to its strategic area, for which a robust association with academic achievement was established [14]; the cascading feature of the original model would still be retained but reduced to the two elements of strategic EI (i.e., EU→EM). We borrow the label “cascading” to refer to this version of the above-presented mediation model.

Adapting the original cascading model to suit the prediction of academic achievement, our next step was to consider which aspects of intelligence and personality to incorporate, as well as which paths from them to the criterion. Concerning intelligence, Joseph and Newman argued that it is the “knowledge-related component of cognitive ability” that primarily affects job performance [30] (p. 59). In the case of school achievement, meta-analytic findings point to verbal ability as a particularly strong predictor [16]; thus, the “cognitive ability” variable from Joseph and Newman’s model would here translate to verbal intelligence (VI). The original cascading model further postulates that intelligence works through the EU branch to indirectly affect job performance (besides having a direct effect on the criterion). The same path may arguably be incorporated in the “cascading” model predicting school achievement: EU involves labeling emotions and propositional thinking with emotional information [33], which may indeed be influenced by overall VI (i.e., VI→EU). This would also be empirically backed by the fact that, of the four EI branches, EU is the strongest correlate of verbal/crystallized intelligence [26,34].

As for personality, Joseph and Newman’ model includes only two of the Big Five, namely C and N. Certainly, C should feature in any model predicting academic achievement, as it is the trait that shows the largest and most robust associations with school performance [22]. Beyond C, at least two other traits should be incorporated in the model, namely O and A [22]. Admittedly, their associations with academic achievement are generally weaker than for C and tend to diminish when other variables (e.g., intelligence) are brought into play. Nevertheless, as both O and A commonly exhibit nontrivial correlations with strategic EI [26,34], this would support the assumption that their effects on academic achievement are at least partially mediated by EI abilities. Regarding these indirect effects in the “cascading” version of the model, we speculated that O is most likely to work through EU. This is because O entails traits such as intellectual curiosity, liberalism, imagination, and an interest in culture [35], which is likely to bring about learning opportunities that, apart from promoting achievement directly, may also spur the development of cognitive abilities [36], including EU. Concretely, greater exposure to emotionally diverse experiences (O) would lead to gains in emotion-related knowledge and reasoning (EU); this, in turn, would also widen one’s spectrum of emotion management strategies (i.e., O→EU→EM), and ultimately enhance academic performance. Concerning the indirect effects of A, we assumed that this trait is most likely to act through the EM branch. More specifically, the tendencies implicated by A, such as altruism, compliance, truthfulness, modesty, tendermindedness, and trustworthiness [35], might orient a person toward envisioning and mentally exploring more constructive and socially desirable emotion regulation strategies, thus enhancing one’s EM skills (i.e., A→EM); these could then be applied to achieve better outcomes at school. Finally, for C, we only assumed a direct effect on academic achievement, but not one that would be mediated by EI. The reason for this is, first, that C has generally exhibited only trivial or non-significant correlations with EI [34], and second, that a prior study found it to predict academic outcomes independently of EM [11]. Incidentally, in Joseph and Newman’s model, C is assumed to have a direct effect on job performance and on EP, but not on either of the two strategic EI branches included in the present model.

In sum, the herein proposed mediation model assumes that VI, C, O, and A directly contribute to academic achievement, with all of these predictors except C also working to enhance school performance through strategic EI. The somewhat more complex “cascading” version of this model proposes that the indirect effects occur along the following paths: from VI to EU to EM, from O to EU to EM, from A to EM, and—in all three cases—ultimately to academic achievement.

### 2.2. Direct Effects Model

The second plausible model of the interaction between EI, intelligence, and personality in predicting academic achievement challenges the two main assumptions of Joseph and Newman’s cascading model and the mediation model in general. First, recall that separate mechanisms were proposed by which EU and EM may affect academic achievement (see Section 1.3). In light of this, it is conceivable that EU does not predict school performance via EM, but that it does so directly. While Joseph and Newman’s and Nguyen et al.’s [30,32] studies clearly showed that the effect of EP on EM is largely/fully mediated by EU (by also estimating the direct path from EP to EM), they did not provide the same type of evidence for the proposed mediation by EM of the relationship between EU and job performance (i.e., a direct path from EU to the criterion was not checked). Judging from MacCann et al.’s study [14], which demonstrated that both EU and EM “are active ingredients in the prediction of academic performance” [14] (p. 169), a direct effect of EU on academic performance might even be more probable than an EU→EM cascade.

Second, regarding Joseph and Newman’s assumption that intelligence and personality are “important antecedents of the EI processes” (p. 69) in predicting job performance [30]—and the extension of this idea to the prediction of academic achievement, as proposed in the mediation model—one might argue that EI’s relationship to intelligence and personality is rather a bidirectional one. For instance, it is conceivable that those who are better at understanding emotions will also be more open to new experiences and the feelings related to them (O) because they have the cognitive tools to process these experiences. Furthermore, those who have at their disposal a large repertoire of emotion regulation strategies and can judge the effectiveness of particular actions in a given situation (high EM) will probably tend to resort to these strategies to avoid or resolve conflicts and thus come across as more affable (A). Moreover, evidence has accumulated in support of the notion that EI is a second-stratum ability, roughly equivalent to other broad intelligence factors appearing at this level of the Cattell–Horn–Carroll (CHC) structure of cognitive abilities [37,38].

All in all, it seems reasonable to assume that a model in which EI is not placed “after” personality and (verbal) intelligence but “alongside” them could equally well explain the targeted outcome and give a more appropriate estimate of the contribution of strategic EI to academic achievement (Figure 1b). After all, MacCann et al.’s [14] meta-analysis showed that overall EI does feature as one of the three most important predictors of academic achievement (along with intelligence and C) and that its strategic branches (entered separately as predictors) even outweigh C in predicting the given criterion.

### 2.3. Moderation Model

Finally, although this possibility has hitherto received little attention in the EI literature, it is quite plausible that EI, intelligence, and personality could be positioned in the same way as are the ability and personality factors of performance in so-called models of talent development, e.g., [39,40]. Specifically, these models propose that ability variables, such as intellectual or social abilities, are directly invested in the process of competence development and ultimately reflected in the level of achievement; personality variables, on the other hand, are assumed to act as moderators, which do not directly affect learning outcomes, but may enhance or stifle the learning process, i.e., the transformation of abilities into competencies and achievement. In our case, this would mean that VI and strategic EI (or each strategic branch) would be the actual predictors of academic achievement, whereas personality would simply moderate their effects on the criterion (Figure 1c). This model would be able to capture the commonly encountered situation in which a student has high VI, but her lack of self-discipline or achievement motivation (low C) is interfering with the learning process such that she ends up underachieving at school [41]. The proposed influence of C on the relationship between intelligence and GPA has already received some empirical support in previous research [42].

Moreover, the “moderation model” would also represent the possible scenario in which certain personality traits decide whether EI will indeed contribute to academic achievement in the ways proposed in the literature and described in Section 1.3. For example, a student might possess the relevant EI abilities to analyze and grasp the emotional meaning of content taught at school (high EU), or to select an effective approach to get the teacher to explain the content to her (high EM), but may also be quite defiant and rebellious (low A), making her reluctant to use her EI skills for the sake of achievement. That personality traits can act as moderators when it comes to the expression of EI was empirically demonstrated by Freudenthaler and Neubauer [43]. Specifically, these authors found the typical emotion management behavior of individuals with higher A and C to be more aligned with their cognitive ability to deal with their own and other people’s emotions (EM). Transferred to the educational context, this finding would imply that students’ A and C are likely to influence whether their EM abilities will translate into successful emotion regulation at school and during learning, i.e., whether the mechanism will be activated by which EM is supposed to promote academic achievement [27,28]. Further supporting the notion that students “must not only have emotional intelligence, but also must be motivated to use it” (p. 39), a study with undergraduates showed that EI incrementally predicted GPA (over general intelligence), but that its effect on the criterion was moderated by C [44].

## 3. The Present Study

Accumulated findings from three decades of research have confirmed that when considered alongside intelligence and personality, EI can still add to the prediction of academic achievement, particularly through its strategic branches. However, to our knowledge, no previous studies have directly investigated how (strategic) EI interacts with intelligence and personality to predict academic achievement. As argued above, several models involving these three clusters of predictors are theoretically and empirically defensible. The present study thus examined the relationship between strategic EI and academic achievement while also considering intelligence and personality, with the main aim of testing the above-proposed models of interaction between the given predictor variables.

We set off with some clear expectations regarding the linear associations between our study variables. First, we expected that strategic EI and its branches would be positively associated with academic achievement, and that, consistent with previous findings, the same would be true for VI and the Big Five traits C, O, and A. Second, we hoped to replicate previous findings regarding the pattern and size of associations between EI, VI, and the Big Five, meaning that strategic EI and its branches should have a moderate positive association with VI, small-to-moderate associations with O and A, and not be significantly related to the remaining three Big Five dimensions. Moreover, we expected that strategic EI would incrementally predict academic performance over VI and the Big Five.

With respect to our main research goal, we hope to have provided a clear rationale for why each of the models described in the previous sections might be expected to receive some empirical support. To sum up, the mediation model assumed that strategic EI would partially mediate the effects of VI, O, and A on academic achievement. A more elaborate, “cascading,” version of this model further proposed that the indirect paths from VI and O would go through EU, cascading to EM, and ultimately to academic achievement, while the effect of A on the criterion would only be partially mediated by EM. In the direct effects model, strategic EI or its two branches were positioned at the same level as VI and the three personality traits (C, O, A), and it was assumed that the EI variables would have a significant independent effect on academic achievement. Finally, the moderator model assumed that the Big Five personality traits, or some of them, might moderate the effects of VI and strategic EI on academic performance. As all of the proposed models are theoretically plausible, they were regarded as competing and the study was designed so as to empirically test their appropriateness.

Numerous studies, including two recent meta-analyses [14,45], suggest that the observed associations between EI and academic achievement can be affected by students’ age and gender. Thus, in examining the proposed models, we also took these variables into account.

## 4. Materials and Methods

### 4.1. Participants

The study sample included 227 students (146 female), whose ages ranged from 13 to 19 years (*M* = 16.50 years; *SD* = 1.42 years). At the time of the study, 108 (47.6%) participants attended compulsory secondary education (from 13 to 16 years old), while 119 (52.4%) received vocational training or baccalaureate (from 17 to 19 years old) in 64 different educational centers in the province of Cádiz (Spain).

### 4.2. Measures

#### 4.2.1. Verbal Intelligence

Verbal intelligence was assessed using the 39-item verbal analogies test ALF7 [46]. The ALF7 is a standard test of verbal analogical reasoning in which respondents must identify one option among four alternatives that adequately complements the presented word based on the relationship established among the first word pair, e.g., *“Day is to night, as white is to… (red, black, clear, clean).”* Each item has a single correct answer that receives one point, thus resulting in an overall score ranging from 0 to 39. Results of prior studies confirmed that the ALF7 is a highly reliable and valid measure of intelligence [46]. For this study, test items were adapted from Serbian to Spanish using the double-translation procedure [47].

#### 4.2.2. Big Five Personality Traits

The Ten-Item Personality Inventory (TIPI) [48] was used to assess the Big Five personality dimensions. Each Big Five dimension is represented by two items, one of which is phrased to present the dimension’s positive pole, and the other its negative pole. Items are scored on a Likert-type scale ranging from 1 (strongly disagree) to 7 (strongly agree). Validation of the Spanish version of the inventory resulted in the conclusion that, apart from shortcomings related to internal consistency, the TIPI is a promising instrument when assessment brevity is a priority [49].

#### 4.2.3. Emotional Intelligence

Emotional intelligence was assessed using the Spanish adaptation of the Mayer–Salovey–Caruso Emotional Intelligence Test (MSCEIT) [50]. In this study, four tasks examining the strategic EI area were administered: Blends, Changes, Emotional Management, and Emotional Relations. Depending on the task, participants are asked to respond to questions by either rating the appropriateness of several alternatives on a 5-point Likert-type scale or by choosing the most appropriate alternative among those presented. Responses were scored using the consensus scoring procedure by the publisher (TEA Ediciones), thus yielding EU and EM scores, as well as an overall strategic EI score for each participant. Prior studies showed that the Spanish version of the test mirrors the excellent reliability and validity features of the original [51,52].

#### 4.2.4. Academic Achievement

Academic achievement was operationalized using grade point average (GPA) as reported by the participants. In Spain, positive grades are coded from 5, indicating sufficient, to 10, indicating excellent achievement. For the purpose of this study, grades were converted to a 0–4 scale, with the higher score indicating better school performance. Information on GPA provided by the students was cross-checked with reports made by their parents.

### 4.3. Procedure

Data were collected during the spring quarter of 2020 as part of a larger study relating parental styles to the cognitive, emotional, and social development of adolescents. All participants were recruited through a call that was passed on from the Andalusian Delegation of Education in Cádiz (Spain) to associations of mothers and fathers of students within different educational centers in the province. Due to the pandemic confinement, the response rate of students and parents within a single educational center was rather low, which is why all those who had accepted the call were included in the study without any additional selection procedure. As a result, participating students and parents were dispersed across 64 educational centers within the province of Cádiz. Students who agreed to participate were asked for an informed consent form signed by either parent. In return for participation, parents received a detailed report on their child’s test results. All questionnaires were completed online, except for the intelligence tests that collaborators directly administered to adolescents on Google Meet. For this purpose, six Ph.D. students were trained and coached to administer the tests. The average total administration time for all tests was about 40 min.

### 4.4. Data Analyses

Descriptive statistics, internal consistencies, correlations between study variables, and gender differences were examined using SPSS v20 (IBM, Armonk, NY, USA). The same software was used to perform hierarchical regression analysis that inspected the incremental validity of strategic EI. Using the maximum likelihood estimation method, structural equation modeling (SEM) was performed in AMOS 16 to test the proposed models of interaction between VI, personality, and strategic EI in predicting school achievement. The goodness of model fit was evaluated based on the following indices and thresholds for good model fit [53,54]: (a) chi-square statistic (χ^2^) and its probability > 0.05, (b) CFI (comparative fit index) ≥ 0.95, (c) TLI (Tucker–Lewis index) ≥ 0.95, (d) RMSEA (root mean square error of approximation) < 0.05, and (e) SRMR (standardized root mean square residual) < 0.05. Comparison of the models was based on the AIC (Akaike information criterion) [55] and ECVI (expected cross-validation index) [56], with smaller values indicating a better fit and greater potential for replication. The significance of indirect effects was examined using the bootstrap estimation procedure, with 2000 cases and a 95% confidence interval.

## 5. Results

### 5.1. Descriptive Statistics, Relationships between Variables, and Gender Differences

Table 1 presents the means and standard deviations, as well as score ranges for all examined variables. Variability indices (ranges and SDs) and absolute skewness values suggested that all scores followed an approximately normal distribution.

Alphas were generally adequate and comparable to those previously established for the same instruments, or even higher in the case of three TIPI subscales (Table 1). A negative exception was the poor internal consistency for A and O.

The correlations between study variables were in the expected direction and of the expected size (Table 1). School achievement was positively related to VI, A, C, O, and to strategic EI and its two branches, with most of these correlations being of medium size. Statistically significant positive correlations were also established between VI and strategic EI (overall and at the branch level), as well as between these variables and O. In addition, EM had a small positive correlation with A and a small negative one with N. Finally, age was positively related to VI, strategic EI (and its branches), and O.

The results of ANOVA for gender differences demonstrated that girls scored higher on C (F_(1225)_ = 17.16, *p* < 0.001, d = 0.58), O (F_(1225)_ = 11.90, *p* < 0.001, d = 0.47), EM (F_(1225)_ = 5.76, *p* < 0.05, d = 0.34), and N (F_(1225)_ = 9.74, *p* < 0.01, d = 0.43).

### 5.2. Hierarchical Regression Analyses

To test the incremental validity of strategic EI in the prediction of GPA, a three-step hierarchical regression analysis was performed: age and gender were entered in step 1, followed by VI, C, O, and A in step 2, and rounded off with strategic EI in step 3. The overall model was statistically significant (F_7219_ = 12.12, *p* < 0.001), explaining 26% of the variance in GPA. Entered in the last step, strategic EI added a significant 2% (*p* < 0.05) to the overall variance explained. The order of significant independent predictors of GPA was C (β = 0.32, *p* < 0.001), VI (β = 0.23, *p* < 0.001), O (β = 0.19, *p* < 0.01), age (β = −0.16, *p* < 0.05), and strategic EI (β = 0.15, *p* < 0.05). When SEI was replaced by EU and EM in the last step, the results for the overall model were practically the same (F_8218_ = 10.83, *p* < 0.001, 26% of the variance explained), but of the two branches, only EU emerged as a significant predictor (β = 0.16, *p* < 0.05), accounting for an additional 2% (*p* < 0.05) of the variance in GPA.

### 5.3. Structural Models

In terms of the predictor variables included and their (direct and indirect) effects on GPA, the models tested were as proposed in the Introduction based on the rationale provided there (see Section 2 and Section 3). The correlations between the exogenous variables were modeled according to the results of correlational analyses in the current data set. Furthermore, age and gender were entered as controls in all models to help minimize unrelated effects and improve the robustness and validity of the results.

#### 5.3.1. Mediation and “Cascading” Models

The mediation structural model hypothesized that VI, O, and A have both direct and indirect effects–via strategic EI–on GPA, while C was modeled to have only a direct impact on the dependent variable. Figure 2a displays the standardized parameters for this model. Paths modeled from age and gender were omitted from this and all subsequent figures to ease interpretation.

The results showed that the direct paths from VI, C, O, and strategic EI to GPA were all significant. The outcome of the bootstrap estimation procedure confirmed that the effects of VI and O on GPA were also partially mediated by strategic EI (Table 2). As for A, both its direct and indirect pathways to GPA were nonsignificant. Finally, while VI had a significant direct effect on strategic EI, this was not the case for O and A. Fit indices suggested that the mediation model had an excellent fit, with χ^2^_(3)_ = 1.452, *p* = 0.694; GFI = 0.998; CFI = 1.00; TLI = 1.093; SRMR = 0.016; and RMSEA = 0.000.

Next, we tested the “cascading” model in which strategic EI was replaced by the EU→EM sequence. As in the initial mediation model, direct pathways were entered from all exogenous variables to GPA, with the additional mediation paths via EU for VI and O, and via EM for A.

Again, direct structural path coefficients from VI, C, and O to GPA were statistically significant (Figure 2b). However, the hypothesized indirect effects on GPA were nonsignificant (Table 2), except for the indirect effect of O on EM via EU. As in the mediation model, both direct and indirect pathways modeled from A to GPA were nonsignificant. However, A had a significant direct effect on EM. Standardized coefficients were also significant for the direct effects of VI on EU and of EU on EM, while the pathway from EM to GPA was nonsignificant. Fit indices for the model were as follows: χ^2^_(8)_ = 9.759, *p* = 0.282; GFI = 0.990; CFI = 0.992; TLI = 0.964; SRMR = 0.026; and RMSEA = 0.031. Since the indirect effect of EU on GPA via EM was nonsignificant, the “cascading” model was modified to include a direct pathway from EU to GPA as well. This direct effect was significant (β = 0.16, *p* < 0.05), but moreover, a significant mediated effect through EU on GPA was established for both VI (β = 0.02, 95% CI [0.001, 0.057]) and O (β = 0.02, 95% CI [0.001, 0.056]). The fit was was again excellent, as for the two previous models: χ^2^_(7)_ = 4.063, *p* = 0.773; GFI = 0.996; CFI = 1.00; TLI = 1.070; SRMR = 0.020; and RMSEA = 0.000.

A comparison of the three tested models based on the AIC (67.451, 83.759, and 80.063, respectively) and ECVI (0.298, 0.371, 0.354, respectively) yielded the starting mediation model as the best fitting and most replicable one. When the two variations of the “cascading” model were compared, a significant χ^2^ difference (5.696, *p* < 0.05) indicated that the modified model in which EU influenced GPA not only via EM but also directly was the better-fitting one.

#### 5.3.2. Direct Effects Model

The first tested model hypothesized that the same predictor variables (VI, strategic EI, O, A, and C) all affect GPA directly; correlations between the predictors were modeled as bidirectional when that was suggested by the results of correlational analyses in the current dataset.

The results showed that all entered variables, excluding A, had significant independent effects on GPA (Figure 3a). A comparison of the 95% confidence intervals for standardized coefficients of significant predictors C (95% CI [0.206, 0.441]), VI (95% CI [0.123, 0.330]), O (95% CI [0.059, 0.328]), and strategic EI (95% CI [0.031, 0.268]) revealed significant overlaps, suggesting that none of them showed supremacy over the others in explaining the variability in GPA. The overall model fit was excellent: χ^2^_(4)_ = 4.968, *p* = 0.291; GFI = 0.995; CFI = 0.994; TLI = 0.956; SRMR = 0.031; and RMSEA = 0.033.

In the second version of the direct effects model, strategic EI was replaced by its branches. Again, A had no significant effect on GPA (Figure 3b). Another important result was that the standardized regression coefficient for the EM→GPA also failed to reach a statistically significant value. On the other hand, the effect of EU on GPA was significant and independent of VI, C, and O. Once more, the 95% confidence interval for the EU effect (95% CI [0.040, 0.276]) overlapped with the lower–upper bound ranges of all other significant predictors: C (95% CI [0.217, 0.445]), VI (95% CI [0.120, 0.327]), and O (95% CI [0.057, 0.326]). The model fit indices suggested an acceptable fit to the observed data: χ^2^_(6)_ = 12.456, *p* = 0.053; GFI = 0.988; CFI = 0.970; TLI = 0.821; SRMR = 0.044; and RMSEA = 0.069.

When A and EM were omitted from the model, its predictive power was slightly improved (R^2^ = 0.27) and so were the model fit indices: χ^2^_(2)_ = 1.436, *p* = 0.488; GFI = 0.998; CFI = 1.000; TLI = 1.041; SRMR = 0.017; and RMSEA = 0.000.

Inspection of the AIC and ECVI values indicated a better fit of the model with strategic EI as a predictor (AIC = 68.968, ECVI = 0.305) compared to the model that included its branches (AIC = 90.456, ECVI = 0.400). However, based on the AIC (53.436) and ECVI (0.236), the best-fitting direct effects model was the one in which nonsignificant predictors, namely, EM and A, were omitted.

#### 5.3.3. Moderation Model

Finally, a series of two-way interactions in regression analyses was performed to test the moderating effects of personality traits on the relationship between VI and strategic EI on the one side and GPA on the other. The moderating effects of C, O, and A were tested in three separate models. Before the analyses, all independents were standardized and product variables were created.

At the start, each model included VI and strategic EI, one of the Big Five traits (C, O, or A), and interaction terms of that particular trait with the other two predictors. Since all exogenous variables were allowed to covariate, all models were initially saturated and the model fit was not obtained. Regardless of that, moderating effects were nonsignificant for all three inspected personality traits, which is why the moderation models were not further tested with EU and EM instead of overall strategic EI.

## 6. Discussion

The current study built on recent meta-analytic findings showing that EI, particularly its strategic area, plays an important role in predicting academic achievement, next to intelligence and the Big Five (most notably C). However, our study took this strain of research a step further by considering and empirically testing several plausible models of the interplay between strategic EI and relevant aspects of intelligence and personality in predicting this outcome. Specifically, we proposed three different models whereby (a) strategic EI partially mediates the effects of VI and personality on academic performance (the mediation model), (b) strategic EI directly predicts academic performance alongside VI and personality (the direct effects model), or (c) the effects of strategic EI and VI on academic performance are moderated by personality (the moderation model). Each model also had a version in which strategic EI was unpacked into its branches, namely, EU and EM.

By testing the proposed models, the current study shed light on several important issues. The first one concerns the position of EI in relation to intelligence and personality as well-established predictors of academic achievement: Is EI a partial mediator of their effects or another independent predictor of this outcome? The second one pertains to the order of the two strategic branches, namely, EU and EM, in predicting (academic) performance: Is there a causal sequence (EU→EM→GPA), as proposed in Joseph and Newman’s cascading model [30,32], or rather two parallel paths (EU→GPA, EM→GPA), as recently suggested by MacCann et al. [14]? The final question refers to the possible interactions between ability variables and personality in explaining academic achievement: Are the effects of VI and strategic EI moderated by personality, as would be implied by talent development models [39,40]?

Before turning to these issues, we shall briefly comment on the linear associations between study variables, and the predictor variables’ contribution to explaining the criterion in question. In this regard, it may generally be noted that the current results largely mirror those from previous studies and meta-analyses, thus supporting the expectations uttered in Section 3.

First, strategic EI and both its branches were positively related to GPA. The strength of the association for EU (r = 0.20, 95% CI [0.08, 0.32]) was slightly lower than would be expected from meta-analytical findings (95% CI [0.28, 0.43]), while for EM (r = 0.17, 95% CI [0.04, 0.29]), it was right within the expected range (95% CI [0.16, 0.35]) [14]. As expected, VI, O, A, and C also emerged as positive correlates of GPA, with the association with the latter being largest for C, which is in accordance with some meta-analytic findings [22].

Second, the pattern of associations between EI, VI, and the Big Five was fully in line with those previously reported. Strategic EI (or at least one of its branches) had a positive correlation with VI, O, and A, and a negative one with N. The correlations were expectedly largest for VI and O, while A was significantly associated only with EM (cf. [34]).

In addition, testing for gender and age differences in strategic EI, we were able to replicate the two most common findings within younger age cohorts; for a recent discussion on age specific demographic differences in EI, see [57]. The first one was that EI increased with age; the second was that girls tended to outperform boys, although, in this study, they did so only on EM.

Finally, in a hierarchical regression analysis, strategic EI (more precisely, its EU branch) accounted for a modest but statistically significant 2% of the variance in GPA above VI and personality after considering age and gender. Thus, our results provide further evidence of the incremental validity of EI above traditional dispositional predictors of academic achievement [14,28,58] and align with previous findings pointing to EU as the EI branch that best predicts school performance [58,59].

### 6.1. Is Strategic EI a Partial Mediator for VI and Personality or an Independent Predictor of Academic Achievement?

Turning to the first of the above-enumerated issues (EI as a mediator or another independent predictor), it is pertinent to discuss the SEM results for the mediation model vs. the direct effects model. The primary thing to notice here is that the first and best-fitting versions of each model, i.e., those including overall strategic EI (Figure 2a and Figure 3a), both had excellent fit indices. This would seem to suggest that either of these two models could be validly representing the interaction of strategic EI, VI, and personality (O, A, and C) in predicting academic achievement.

Looking at the results for the mediation model, we find confirmation for the assumption that VI, C, O, and strategic EI directly affect academic achievement, but that the effects of VI and O on GPA are also partially mediated by strategic EI. The present results thus provide initial empirical support for the proposed mechanisms by which verbal ability and intellectual traits (O) work to promote school performance (see Section 2.1). More precisely, apart from contributing directly to learning and achievement (by waking interest in and enhancing the processing of academic contents), both VI and O seem to be implicated in the development of abilities to understand and manage emotions, which are then exercised in attaining favorable outcomes at school.

On the other hand, the strong fit obtained for the direct effects model implies that strategic EI does not merely act as a mediator through which VI and O affected academic achievement but is a relevant ingredient of school success in its own right. In fact, a comparison of the 95% CIs for the direct and total (i.e., direct + indirect) effects of VI (β_VIdir_ 95% CI [0.12, 0.33], β_VItot_ 95% CI [0.14, 0.35]) and O (β_Odir_ 95% CI [0.06, 0.33], β_Otol_ 95% CI [0.08, 0.34]) in the mediation model revealed a full overlap, which indicates that the introduction of indirect pathways via strategic EI to GPA did not substantially improve the explanatory power of VI and O. Thus, in terms of parsimony, and given that some direct (O→strategic EI) and indirect pathways (A→strategic EI→GPA) in the mediation model were non-significant, the direct effects model seems to outweigh the former. In addition, the direct effects model is also more compatible with the empirically supported proposition that EI is a broad intelligence factor, placed at the same level of the CHC hierarchy as other broad abilities, such as verbal/crystallized intelligence [37,38]. Finally, it also accords with MacCann et al.’s [14] remark that the effect of EI, and particularly of EU, on academic performance “cannot be explained solely by its overlap with intelligence” (p. 167).

### 6.2. Do EU and EM Work in Sequence or in Parallel to Predict Academic Achievement?

We now turn to a consideration of the order of the two strategic branches, namely, EU and EM, in predicting academic achievement. A necessary starting point is to evaluate the model versions (of both the mediation and the direct effects model) that resulted from substituting overall strategic EI with its branches (Figure 2b and Figure 3b) In both cases, the resulting models had good fit indices but were nevertheless outperformed by the corresponding simpler version that included only the aggregate strategic EI score. The most probable reason for this is the lack of specific criterion variance explained by EM, as shown by its nonsignificant contribution to GPA when combined with other predictor variables. Thus, the effect of strategic EI in the first versions of both models practically comes down to the effect of EU, leaving EM as a redundant variable in the second model versions. What does this mean for the proposed sequential order of the two EI branches in predicting academic achievement?

First of all, the current data do not support the proposition that EU works through EM to enhance school performance. In fact, the introduction of a direct pathway from EU to GPA within the “cascading” model significantly improved the model fit indices and confirmed that the effect of EU on the criterion was exclusively a direct one. This finding contradicts one of the basic assumptions of Joseph and Newman’s cascading model, namely, that there is a causal chain in which EU precedes EM, and the latter serves as the direct link to (job) performance [30]. Two reasons may explain why our study reached a different finding than that presented by Joseph and Newman [30] and Nguyen et al. [32]. The first and most obvious explanation draws on the difference in the criteria predicted: while the cascading model of EI seems to fit when predicting job performance, it evidently does less so when the criterion is academic achievement. This might ultimately be related to the fact that, unlike for job performance, a broad and solid mechanism is conceivable by which EU can directly affect achievement in any academic program involving language arts and humanities. As for the other possible explanation, recall that neither Joseph and Newman nor Nguyen et al. reported the fit indices and standardized estimates for any alternative models that included a direct path from EU (or EP for that matter) to job performance. Consequently, it remains dubious whether the cascading model of EI really provided the best fit to the data in the two respective studies. In other words, although the present findings apparently contradict those reported in the referred-to studies, we found upon closer inspection that the results taken to support the cascading model were not conclusive enough in the first place. Overall, then, it is not surprising that we found EU to have a direct effect on GPA rather than one that is mediated by EM, especially since EU was recently judged to play a “critical role” in academic performance [14] (p. 169). The question remains, however, why EM did not contribute to predicting the criterion in the present study, while it did so in MacCann et al.’s meta-analysis [14]. A tentative answer is provided below, as we consider the mechanisms by which EU and EM affect academic achievement.

While the present study clearly speaks against a cascade from EU to EM to academic achievement and in favor of EU having a direct effect on the outcome (comparable to the effects of VI and O), it does not provide any direct insights into the mechanisms by which this effect was exerted. Nevertheless, the fact that EU incrementally predicted GPA and that its effect on the latter was not mediated by EM resonates with the proposition that EU contributes to school achievement in its own right, namely, as a resource for mastering those aspects of the curriculum that require an understanding of human emotions and emotionally driven (inter)actions, as is the case in the language arts and humanities [14,60]. As for EM, it should be noted that the mechanisms by which this EI branch was hypothesized to affect school achievement (see Section 1.3) are somewhat less direct: Unlike emotional vocabulary and understanding, which may be directly invested in mastering certain academic contents, knowledge about effective ways to regulate emotions (i.e., EM as operationalized by EI tests) first has to be translated into actual emotion management, and then there might be other intermediate steps—e.g., establishing good social relations at school [27,28]—before EM abilities can be reflected in learning and, finally, achievement. Taking this truly strategic approach to learning, or even just turning one’s emotion management knowledge into actual control over emotions in the academic context, might be particularly challenging for adolescents, who—more than other age groups—are likely to act on their impulses despite knowing better [61]. In any case, it seems to us that, compared to EU, EM might be the more distal and thus also less stable predictor of school achievement, which also explains its non-significant contribution in the present sample.

### 6.3. Do Personality Traits Moderate the Effects of Abilities or Contribute Alongside Them?

Finally, regarding the third issue, i.e., the role of personality traits in a model that serves to predict academic achievement, the results of the present study were quite clear: The proposed moderating effects of personality on the relationship between VI and strategic EI, on the one side, and academic performance, on the other, were not borne out by our data. Given the excellent fit of the other two models (i.e., mediation and direct effects), whereby personality traits were assumed to directly and/or indirectly contribute to academic achievement, it seems fair to conclude that their role is not merely one of “catalysts” that enhance or inhibit the learning process, the outcomes of which are basically influenced by students’ level of abilities [39], be they verbal, emotional, or of another type. Certainly, some studies have provided empirical support for the proposition that personality traits moderate the expression of abilities—in this case, EI [43]—and we are not suggesting that this is not one viable path by which personality may affect academic performance. However, it is obvious from our data that a model assuming that this is the only way in which personality is implicated in academic achievement is overly simplistic.

According to the present results, two personality traits, namely, C and O, contributed independently to adolescents’ school achievement, with C also surfacing as the strongest in this particular set of predictors. It is thus clear that personality, and especially C, remains a crucial ingredient of academic performance, even when increasing content complexity is expected to put more weight on students’ intellectual capacities (cf. [19]). In fact, personality may affect achievement via paths beyond those proposed and tested in the present study, which remains to be tackled by future research.

### 6.4. Practical Implications

Providing further evidence of the importance of strategic EI and the “critical role” of EU [14] for adolescents’ academic achievement, the present study also justifies the efforts placed into developing theory-driven approaches to enhance students’ abilities to label and understand emotions (e.g., RULER) [62]. More specifically, it suggests that interventions designed to promote students’ emotional vocabulary and understanding of “feeling words” might be an effective means to improve their academic performance in areas that require the use and understanding of emotional language (i.e., the language arts and humanities).

While EM did not contribute to academic performance in the present sample, we do not take this result as undermining the importance of this EI branch; rather, we speculate that it might be challenging, particularly for adolescents, to draw upon their emotion regulation knowledge to actually manage their own and others’ emotions, and ultimately perform better in terms of grades. From this point of view, it would also make sense to employ interventions that teach students how to handle and respond to school situations in an emotionally intelligent rather than impulsive manner. Accumulating evidence testifies to the usefulness of such tools for improving individual student outcomes [63] and the quality of learning environments [64]. Although important at all educational levels, these systematic interventions might be especially fruitful for adolescents, supplying them with the skills necessary to deal with the overwhelming emotional experiences typical of that age and preparing them for the complex social environments of adulthood.

### 6.5. Limitations and Future Directions

Before concluding, we would like to acknowledge several methodological limitations of the present study.

First, our findings are based on cross-sectional data and, thus, all causal relations tested and confirmed via SEM should be taken with the necessary caution required by virtue of the study design. As is usually the case with modeling the presumably complex interplay of factors that determine an outcome such as academic achievement, only (multiple) longitudinal studies would allow for more definite conclusions about the dynamics of all relevant dispositions.

Second, our study included participants of a limited age range and, thus, the current findings may not fully generalize to other age groups. The relative contribution of different dispositional traits in the prediction of academic achievement can change with age, and the same could be the case for EI. Apart from that, it should also be acknowledged that the current study did not control for the possible effects that other demographic variables, particularly SES, could have on the established associations.

Next, certain limitations were related to the measures used to assess some of the main study variables. Although we employed a well-validated test of VI, a more reliable assessment of this, or any other, broad ability would require at least two to three separate test indicators, which is something that we certainly recommend for any future replications of the present study. Moreover, for the sake of brevity, we opted for a very brief, 10-item measure of the Big Five, which compromised the internal consistencies for some traits. Even with such low reliabilities, the Big Five exhibited the expected pattern of relationships with other study variables, which speaks in favor of the robustness of these associations. Nevertheless, any future investigation of the same research question should preferably rely on a longer and more reliable Big Five inventory. Finally, given the known legal difficulties in obtaining official school records, we had participants report their GPA, which was then cross-checked only via parental reports on the same variable. Certainly, it would have been optimal to collect the data on GPA from the schools directly, though meta-analytic findings reveal a rather strong correlation (ρ = 0.82) between self-reported and official high school GPA [65].

Not a limitation of the present study, but rather an incentive for future ones, would be the need to chart out how different educational outcomes are predicted by particular EI branches, i.e., wherein the predictive power of each branch lies. A possibility suggested by the present study, in conjunction with some previous research, is that EU might be the branch that is most directly involved in predicting school performance in terms of grades [14], while EM might play a more important role vis-à-vis outcomes such as social adjustment and students’ well-being at school (cf. [27,28,66]).

## 7. Conclusions

To our knowledge, this was the first study to propose and empirically test several plausible models of the interplay between strategic EI, VI, and personality in predicting academic achievement. To begin with, our results confirmed that strategic EI is positively associated with and incrementally predicts academic achievement: in this case, it was overall strategic EI and its EU branch that predicted GPA over VI and the Big Five traits of O, A, and C. Moreover, by using SEM to explore several possible arrangements in which the above predictors might affect academic achievement, we were able to draw the following conclusions: First, while they certainly allow for the possibility that the effects of VI and O on academic achievement are partially mediated by strategic EI, overall, our results make a better case for a parsimonious “direct effects model,” whereby strategic EI is placed alongside VI and the relevant personality traits (i.e., O, A, and C) as another independent predictor of GPA. Second, from the present results, it seems unlikely that there is a cascading effect of EU on EM and thus on academic achievement, but rather that EU affects the criterion independently of EM (which, in this sample, did not even contribute at a statistically significant level to predicting GPA). Assuming that in other instances, both EU and EM can contribute to academic performance (as suggested by prior research), the present findings would imply that they are likely to do so in a parallel rather than in a sequential order. In fact, our results indicate that—among the two strategic EI branches—it is EU, not EM, that is the more direct and “proximal” predictor of academic achievement. Finally, our study suggests that personality traits do not merely moderate the effects of abilities, i.e., VI and strategic EI, on academic achievement, but that some of them, specifically C and O, substantially affect this outcome independently of VI and EI and, in the case of O, partially through EU abilities.

## Figures and Tables

**Figure 1 ijerph-18-13166-f001:**
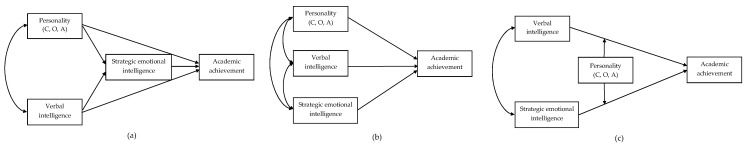
Theoretical models of the interaction between strategic EI, verbal intelligence, and personality in predicting academic achievement: (**a**) mediation model; (**b**) direct effects model; (**c**) moderation model. Note. In model (**a**), C is assumed to have only a direct effect on academic achievement, while for O and A, both direct and indirect effects are hypothesized.

**Figure 2 ijerph-18-13166-f002:**
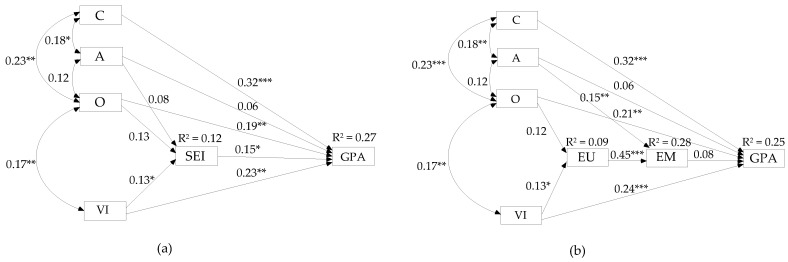
Standardized parameter estimates for the (**a**) mediation and (**b**) “cascading” models. GPA, grade point average; VI, verbal intelligence; A, agreeableness; C, conscientiousness; O, openness; SEI, strategic emotional intelligence; EU, emotion understanding; EM, emotion management. * *p* < 0.05, ** *p* < 0.01, *** *p* < 0.001.

**Figure 3 ijerph-18-13166-f003:**
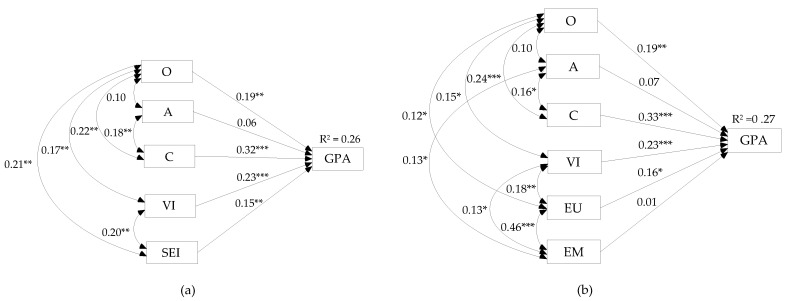
Standardized parameter estimates for the direct effects model with (**a**) strategic EI and (**b**) emotion understanding and management branches. GPA, grade point average; VI, verbal intelligence; A, agreeableness; C, conscientiousness; O, openness; SEI, strategic emotional intelligence; EU, emotion understanding; EM, emotion management. * *p* < 0.05, ** *p* < 0.01, *** *p* < 0.001.

**Table 1 ijerph-18-13166-t001:** Descriptive statistics, internal consistencies, and correlations between variables.

Variables	α	M (SD)	Min.–Max.	Skew	1	2	3	4	5	6	7	8	9	10
**1. GPA**	-	2.68 (1.01)	1.00–4.00	−0.34	-									
**2. VI**	0.84	49.44 (14.45)	20.59–87.21	0.36	0.29 **	-								
**3. E**	0.59	4.63 (1.43)	1.00–7.00	−0.21	−0.01	−0.12	-							
**4. A**	0.30	5.26 (1.01)	2.50–7.00	−0.38	0.15 *	0.03	−0.11	-						
**5. C**	0.60	4.98 (1.46)	1.00–7.00	−0.31	0.38 **	0.08	0.03	0.18 **	-					
**6. N**	0.68	4.13 (1.47)	1.00–7.00	0.04	0.03	−0.04	0.10	−0.44 **	0.04	-				
**7. O**	0.18	5.37 (1.21)	1.50–7.00	−0.42	0.30 **	0.19 **	0.11	0.13	0.24 **	−0.01	-			
**8. EU**	0.86	94.06 (14.08)	56.67–133.94	−0.06	0.20 **	0.19 **	0.05	0.04	−0.01	−0.04	0.19 **	-		
**9. EM**	0.80	96.55 (13.83)	65.48–130.96	−0.10	0.17 *	0.16 *	0.04	0.17 *	0.12	−0.16 *	0.19 **	0.48 **	-	
**10. SEI**	0.85	94.40 (14.06)	57.95–130.31	−0.02	0.22 **	0.20 **	0.06	0.12	0.07	−0.12	0.23 **	0.87 **	0.83 **	-
**11. Age**	-	16.50 (1.42)	13–19	-	−0.01	0.19 **	−0.09	0.10	0.03	−0.10	0.24 **	0.24 **	0.18 **	0.25 **

Note: GPA, grade point average; VI, verbal intelligence; E, extraversion; A, agreeableness; C, conscientiousness; N, neuroticism; O, openness; EU, emotion understanding; EM, emotion management; SEI, strategic emotional intelligence. ** *p* < 0.01; * *p* < 0.05.

**Table 2 ijerph-18-13166-t002:** Standardized indirect effects and 95% confidence intervals for the mediation and “cascading” models.

Model Tested	Model Pathways	Estimated Effect	95% CI	
			Lower	Upper
Mediation model	A→strategic EI→GPA	0.01	−0.003	0.046
	O→strategic EI→GPA	0.02 *	0.001	0.054
	VI→strategic EI→GPA	0.02 *	0.002	0.056
“Cascading” model	A→EM→GPA	0.01	−0.003	0.042
	O→EU→EM	0.06 *	0.001	0.125
	VI→EU→EM	0.06	−0.002	0.122
	EU→EM→GPA	0.04	−0.014	0.093
	O→EM→EM→GPA	0.01	−0.001	0.020
	VI→EU→EM→GPA	0.01	−0.001	0.019

* *p* < 0.05.

## Data Availability

The database is available from the corresponding authors upon reasonable request.

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
