# Peer review of "The Role of Strategic Emotional Intelligence in Predicting Adolescents’ Academic Achievement: Possible Interplays with Verbal Intelligence and Personality"

_ijerph, 2021, doi:10.3390/ijerph182413166_

Round 1

Reviewer 1 Report

The article as a whole is well-established, and the subject is interesting. The following comments may assist in improving the article:

The authors should ask the help of native English-speaking proofreader, because there are some minor linguistic mistakes that should be fixed.

  • More suitable title should be selected for the article.
  • It is proposed that the article be supplemented with a flowchart illustrating the research technique.

Reviewer 2 Report

I appreciate the opportunity to review your paper.  Your paper covers a timely and important topic.  With that said, I do have some significant concerns and some suggestions, as outlined below.

My primary concern is that you are testing moderation and mediation hypotheses using cross sectional data.  I see that you recognize this in your limitations section, but I believe you need to consider additional ways to mitigate the risks of common method bias in order for your results to be meaningful.  For example, would it be possible to validate the participant’s self-reported GPA with their school.  Your population is not so large to make this an unreasonable task.  Students would need to sign a release, of course, but it appears that you may be able to do this.  I strongly encourage you to do so, as this is a serious foundational concern that challenges your findings.

Second, I recognize that you are “exploring” the potential for interplay between your constructs, but it would be helpful if you could find some additional extant studies, beyond the ones you have included (i.e., Joseph & Newman, Freudenthaler and Neubauer, Gagne, and Heller et al) to substantiate your models.

You rely heavily on the Joseph & Newman article as a base for you Mediation hypotheses.  As you note, Joseph & Newman were studying job performance.  Can you find any other extant literature that might help you substantiate your mediation hypotheses relevant to your study constructs?  Without this, I think you may need to substantiate these hypotheses a bit more with some theoretical bases.

I have more serious concerns about your limited substantiation for your posed moderation model.  In your opening sentence (line 260), you use the phrase “viable possibility.”  Being honest, this feels a bit like you are “searching” for relevance/significance -  rather than a building a robust, theory and empirically supported study.  Again, the extant research you rely on does not pertain to academic performance but rather talent development and mental ability to deal with their own and other people’s emotions. 

The MacCann et al study seems to mirror your posed direct effects model, and your ultimate proposed best model.  With that said, it would be helpful for you to say a bit more about how your findings are building on MacCann’s findings or are adding more empirical value to the literature.

What about additional control variables?  For example, socio-economic class, child order, family status, etc.  It seems to me that there are many other variables that need to be included, beyond age and gender, to help control for noise.  I have not taken the time to do this, but a review of the literature on academic achievement would be useful in identifying additional important control variables.

Personality – You note that your study/model explores “C,” “O,” and “A”, but you include outcomes for (N”.  I would recommend that you remove reference to factors not included in your model.  For example, you don’t even mention “N” until line 394.  Best, in my judgment, to leave it out of your results and discussion.  Alternatively, you could consider including N in your model or putting references to “N” in your discussion or future research if there are valuable things to draw from these findings.

Lines 395-398:  Not sure why you have included this since you did not pose related hypotheses about gender.  Again as noted above, this might be valuable in your discussion or future research section.

Table 2.  I note that the dependent variable for the mediation pertaining to VI – SEI is “AA” rather than “GPA.”  Does this actually reflect a different DV used here, or possibly a typographical error?

The outline of your models (between lines 167 and 168):  Model (a) shows Personality (C, O, A) together – all as having direct and indirect effects.  This should be amended to reflect your posed relationships in which C is hypothesized to only have direct effects.  Displayed as you have it created some confusion as I was trying to reconcile your models with your narrative.

Lines 222/223:  Narrative here is unclear.  Not sure what you are conveying.

Lines 453-459:  your exploration of potential direct effects of EU here, similar to my comment above, come across as your searching for significance – and can lend to undermine the credibility of your findings.  You have not identified this relationship in your hypotheses, so I would recommend you remove them from findings.  It might make more sense for you to include them in some post hoc comments in the discussion or future research. 

Best wishes as you continue to work with these data.

Reviewer 3 Report

This study is focused on the relationships among adolescents' academic achievement, strategic emotional intelligence, verbal intelligence, and personality. An interesting aim was to analyze the possible direct, mediating or moderation role of the strategic emotional intelligence. Findings of this study could be interesting to improve adolescents' academic achievement through the intervention on this variable.

In this study, the main variables are properly defined and explained, and their possible relationships are well justified. The methodological strategy is correct, and all the results are properly described.

In my opinion, the authors should only consider a few minor questions.

Materials and Method.

Participants.

Authors should include more information on the number of students in each educational grade. Also, should be indicated how many students from each educational center participated in this study. It is not clear why only a small number of students in each center participated in this study, when many centers were contacted in this research (64 different educational centers participated in this research). It is not clear the criteria used for the selection of this sample, and why the sample size is small in this study when authors contacted many educational centers.

The main characteristics (public or private, rural or urban, size, ...) of these schools need to be provided.

Procedure.

Authors need to provide more information about the process used to select the educational centers participating in this study.

Limitations

The small size of the sample needs to be considered.

Another minor question:

There is a typo in the manuscript. A parenthesis is missing in line 418:

“… there (see Sections 2 and 3). The correlations between exogenous variables were…” in place of “… there (see Sections 2 and 3 The correlations between exogenous variables were…”.

Round 2

Reviewer 2 Report

First, I want to commend you for the speed at which you were able to consider feedback and provide a substantially revised manuscript.  I appreciate the revisions you have made.  One remaining point, which I think would make your mediation and moderation tests more robust, would be to cross check reported GPA's with the school.  I note that you have cross checked them with parent reports of their child's GPA, but the parental relationship is quite close.  A cross check with an independent source, i.e., the school, would be stronger - and may still be quite viable.

I appreciate the opportunity to review your manuscript.

Author Response

We once more thank you for careful consideration of our manuscript. Clearly, we share your concerns regarding the self-reported GPA, which is why we have crossed checked it with parent reports. Unfortunately, according to current Spanish legislation, it is not possible to check the official GPA of students at their schools. To do so, it would take a lot of paperwork and permissions that would have taken more than a year. At this point we can only add this as a limitation to our findings, which we have done in the revised version of the manuscript.